# LARGE LANGUAGE MODELS PERFORM DIAGNOSTIC REASONING

**Cheng-Kuang Wu**,* **Wei-Lin Chen**,* **Hsin-Hsi Chen**
National Taiwan University, Taiwan
{ckwu,wlchen}@nlg.csie.ntu.edu.tw, hhchen@ntu.edu.tw

## ABSTRACT

We explore the extension of *chain-of-thought* (CoT) prompting to medical reasoning for the task of automatic diagnosis. Motivated by doctors' underlying reasoning process, we present *Diagnostic-Reasoning CoT* (DR-CoT). Empirical results demonstrate that by simply prompting large language models trained only on general text corpus with two DR-CoT exemplars, the diagnostic accuracy improves by 15% comparing to standard prompting. Moreover, the gap reaches a pronounced 18% in out-domain settings. Our findings suggest expert-knowledge reasoning in large language models can be elicited through proper promptings.

## 1 INTRODUCTION

Recently, *Chain-of-Thought* (CoT) prompting Wei et al. (2022) has lunched a line of works on eliciting complex reasoning for large language models (LLMs) (Kojima et al., 2022; Madaan et al., 2022; Chen et al., 2022; Suzgun et al., 2022). However, they mainly focus on arithmetic, commonsense, and symbolic reasoning tasks. In this work, we extend the CoT framework to medical reasoning. Specifically, we investigate the task of automatic diagnosis (AD), which requires knowledge-intensive, multi-step *diagnostic reasoning*.

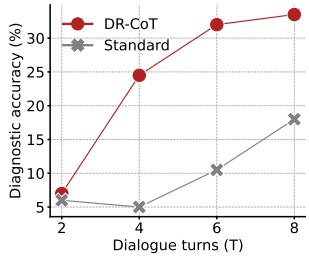

Typical patient-doctor interactions consist of an iterative question-answering (QA) sessions. In each dialogue turn, doctors inquire patients about clinical evidence (e.g., their backgrounds, symptoms experienced), and form a *differential diagnosis* (DDx), a short list of possible diagnoses, in their underlying reasoning (Graves, 2002; Rhoads et al., 2017). The DDx then guides doctors in determining the next questions to ask, and patients' answers provide new evidence for refining the DDx. Once the list is sufficiently narrowed, doctors can establish a final diagnosis. We hypothesize such diagnostic reasoning process—from evidence to DDx to the next question—serves as the hidden CoT for AD.

Figure 1: DR-CoT achieves better diagnostic accuracy and converges more rapidly than standard prompting.

An ideal dialogue system for automatic diagnosis (DSAD) requires the system to interact with patients in free-text natural language. However, prior attempts mainly focus on the dialogue management component of DSAD, which only processes structured information (i.e., takes symptoms as input and output the next symptom to inquire or choose a diagnosis) (Xia et al., 2020; Liao et al., 2020; Liu et al., 2022). Moreover, most existing works use reinforcement learning, and can only predict 4∼5 diagnoses due to the limitation of large action space for symptoms and the lack of holistic datasets (Wei et al., 2018; Xu et al., 2019; Zhao et al., 2021).

In this work, we propose: **(1)** a few-shot LLM-based dialogue system for AD. **(2)** a novel *Diagnostic-Reasoning CoT* (DR-CoT) which elicits reasoning in LLMs towards better AD. **(3)** a language-model-role-playing evaluation framework simulating realistic patient-doctor interactions. To the best of our knowledge, we are the first to introduce LLMs into DSAD, and present a non-pipelined approach which in principle, has an unbounded diagnosis label set. Empirical results show our dialogue system with DR-CoT prompting outperforms standard prompting by a striking 15% on diagnostic accuracy (Figure 1), and the improvement holds for out-domain experiments (Figure 2).

---

*Equal contribution, authors listed alphabetically.

## 2 DR-CoT: DIAGNOSTIC-REASONING CHAIN OF THOUGHT

**Baseline.** We construct our dialogue system based on InstructGPT (Ouyang et al., 2022). Inspired by Few-Shot Bot (Madotto et al., 2021), we prompt the model with the template: "*[I][S][D]*", where I, S, and D denote the instruction, shots, and dialogue history, respectively (we provide an example in Appendix A.1). Each shot in S is a *complete* doctor-patient conversation, and D refers to the dialogue history of the current *incomplete* conversation. Concretely, given $D = \{(q_i, a_i)\}_{i=1,...,t-1}$, where $q_i$ is the utterance generated by our dialogue system, $a_i$ is the utterance from the patient, the output response at turn $t$ is $q_t$. And for the next turn $t+1$, we prompt the model with the identical I, S, and an updated $D = \{(q_i, a_i)\}_{i=1,...,t}$. Once the model deems the gathered evidence is sufficient, it generates a last utterance $d$ to establish the diagnosis and complete the dialogue.

**DR-CoT Prompting.** To design the DR-CoT prompt, we simply apply two modifications comparing to the standard (i.e., baseline) prompt (we provide an example in Appendix A.2): **(1)** Augment I by instructing the model to first summarize the current observed evidence and formulate a DDx, then make inquiries based on the DDx. **(2)** Replace the $\{q_i\}_{i=2,...,T} \in$ S by "*Based on the evidence [E], the ranked differential diagnosis is [DDx]. To narrow down the differential diagnosis, the next question to ask is $q_i$.*", where $T$ is the number of turns and E is the observed evidence in dialogue history. The DDx guides the model to generate a crucial $q_i$ that leads to better evidence gathering, upon which the model compiles a more focused DDx, zooming in on the final diagnosis.

**Evaluation Framework.** Prior works generally evaluate DSAD by interacting with a lookup-table-like patient, which responds to inquiries in a binary, non-natural language fashion. Such setting deviates from real clinical scenarios. For a realistic testbed, we present a language-model-role-playing framework where the LLM plays both sides of the conversation, and performs evaluation by self-chat (Li et al., 2019; Ghandeharioun et al., 2019).

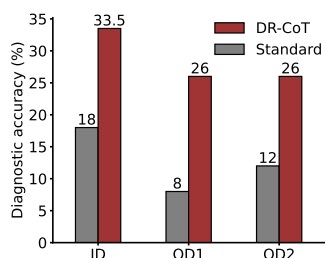

Specifically, after the DSAD generates a $q_i$ at turn $i$, we construct a *patient bot* by prompting LLM with the template: "*[I][S][P][D]*", where P is the patient profile (e.g., background information, symptoms, and medical history) and the last utterance of D is $q_i$ (we provide an example in Appendix A.3). The LLM is instructed to act as a patient and generate a response $a_i$ faithfully, namely, to answer $q_i$ based on the *script* provided in P. [1] We then append $a_i$ to D for prompting our DSAD. In summary, the evaluation is fully automated by prompting LLMs to switch between a doctor role (i.e., the DSAD) and a patient role, and the interaction ends once the DSAD generates $d$.

Figure 2: The improvement on two out-domain testset suggests the elicited diagnostic reasoning ability is agnostic to specific exemplars.

**Results.** We adopt the DDXPlus (Tchango et al., 2022) dataset for our experiments. Using two shots (i.e., exemplars), we conduct experiments with in-domain (ID) setting (the exemplars and test data share the same initial evidence; Figure 1) and out-domain (OD) setting (with different initial evidence; Figure 2). Both settings exhibit substantial performance gain with our DR-CoT prompting. To evaluate the accuracy in each dialogue turn $T$ (Figure 1), if the final diagnosis has not been reached, we instruct the model to make the final diagnosis in turn $T$'s prompt (see in Appendix B.3 for details). In addition, we conduct a small-scale human evaluation (see Appendix C for details and results) supporting our hypothesis that DR-CoT enables LLMs to ask more critical questions for establishing the correct diagnosis. Details of other experimental setups including the DDXPlus dataset and the selection of in-domain and out-domain test sets are provided in Appendix B.

## 3 CONCLUSION

In this work, we propose DR-CoT prompting, which elicits the diagnostic reasoning ability in LLMs with striking empirical improvement, and introduce the first LLM-based DSAD along with a novel language-model-role-playing evaluation framework. Our findings suggest through proper prompting, the potential expert-knowledge reasoning in LLMs can be unlocked for promising applicability.

---

[1] We invite a physician to evaluate 100 sampled $(q_i, a_i)$ pairs, where 95 $a_i$s are considered faithful w.r.t. P.

URM STATEMENT

The authors acknowledge that at least one key author of this work meets the URM criteria of ICLR 2023 Tiny Papers Track.

ETHICS AND REPRODUCIBILITY STATEMENT

In this work, we aim to provide a preliminary exploration into the diagnostic reasoning ability of large language models. Note that the results are not validated in any real-world clinical settings. Though with great potential of assisting clinicians in scenarios like telemedicine services (e.g., the doctors can review the interaction history and associated diagnosis predictions, before making final decisions or perform further inquiries), the current prototype of our dialogue system for automatic diagnosis is not yet directly applicable and requires thorough investigation. To facilitate future research, the source code for conducting the experiments will be provided via a GitHub link in the camera-ready version upon acceptance.

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

OUTLINE OF APPENDIX

We provide additional details in this appendix for our work entitled "*Large Language Models Perform Diagnostic Reasoning*". The content is organized as follows:

- Section A contains examples of standard prompting (i.e., baseline) (Section A.1), DR-CoT prompting (Section A.2), and patient bot prompting (Section A.3).
- Section B contains details of DDXPlus and the selection for in-domain and out-domain samples (Section B.1), the model (Section B.2), and the dialogue turn setting (Section B.3).
- Section C contains human evaluation results for qualitative assessment of DR-CoT.

## A  EXAMPLE PROMPTS

### A.1  AN EXAMPLE OF STANDARD PROMPTING (BASELINE)

```
Act as a doctor taking a medical history from a patient. Try your best to ask comprehensive and important
questions in order to make an accurate final diagnosis. The diagnosis should be selected from the possible
diagnoses listed below.

Possible diagnoses: Bronchiolitis, Pericarditis, Myocarditis, HIV (initial infection), Bronchospasm / acute
asthma exacerbation, Acute pulmonary edema, Pneumonia, Pancreatic neoplasm, Bronchitis, Atrial fibrillation,
Anaphylaxis, Panic attack, Sarcoidosis, Pulmonary neoplasm, Spontaneous rib fracture, Viral pharyngitis, Croup,
Possible NSTEMI / STEMI, Chagas, Cluster headache, Acute dystonic reactions, Scombroid food poisoning, Acute
rhinosinusitis, Anemia, Spontaneous pneumothorax, Guillain-Barré syndrome, Whooping cough, Myasthenia gravis,
SLE, Ebola, Pulmonary embolism, Boerhaave, Unstable angina, Acute otitis media, Stable angina, Inguinal hernia,
Epiglottitis, PSVT, Tuberculosis, Acute laryngitis, Acute COPD exacerbation / infection, Localized edema,
Larygospasm, Influenza, GERD, Bronchiectasis, Allergic sinusitis, Chronic rhinosinusitis, URTI

<History taking>
Doctor: [ask] How may I help you today?
Patient: I have a cough.
Doctor: [ask] What's your sex and age?
Patient: I am an 81-year-old female.
Doctor: [ask] Did you have fever?
Patient: Yes, I have a fever.
Doctor: [ask] Did you have dyspnea or difficulty breathing?
Patient: No.
Doctor: [ask] Do you have sore throat, runny nose, or nasal congestion?
Patient: Yes, I have a sore throat.
Doctor: [ask] What's your body temperature?
Patient: My body temperature is not measured.
Doctor: [ask] Do you have myalgia or muscle soreness?
Patient: Yes, I have diffuse muscle pain.
Doctor: [ask] Do you feel extremely fatigue?
Patient: Yes, I am so tired that I am unable to do my usual activities or I am stuck in my bed all day long.
Doctor: [inform] Based on your symptoms, the most likely diagnosis is Influenza.

<History taking>
Doctor: [ask] How may I help you today?
Patient: I have a cough.
Doctor: [ask] What's your sex and age?
Patient: I am a 24-year-old female.
Doctor: [ask] Do you have shortness of breath?
Patient: No.
Doctor: [ask] Do you have sore throat, runny nose, or nasal congestion?
Patient: Yes, I have a sore throat.
Doctor: [ask] Do you have fever?
Patient: No.
Doctor: [ask] Do you have a burning sensation that starts in your stomach then goes up into your throat, and can
be associated with a bitter taste in your mouth?
Patient: No.
Doctor: [ask] Have you noticed any changes in your voice like hoarseness?
Patient: Yes, I have noticed that the tone of my voice has become deeper, softer or hoarse.
Doctor: [ask] Do you smoke?
Patient: Yes, I smoke cigarettes.
Doctor: [ask] Have you been in contact with people with similar symptoms in the past 2 weeks?
Patient: Yes, I have had a cold in the last 2 weeks.
Doctor: [inform] Based on your description, the most likely diagnosis is Acute laryngitis.

<History taking>
Doctor: [ask] How may I help you today?
Patient: I have a cough.
Doctor: [ask] What's your sex and age?
Patient: I am a 44-year-old male.
Doctor: [ask] Do you have fever?
Patient: No.
Doctor: [ask] Do you have shortness of breath?
Patient: No.
Doctor: [ask] Do you have chest pain or tightness?
Patient: No.
Doctor: [ask] Do you have a productive cough with yellow or green sputum?
Patient: No.
Doctor: [ask] Do you have a history of smoking?
Patient: No.
Doctor: [ask] Have you been in contact with people with similar symptoms in the past 2 weeks?
Patient: No.
Doctor: [ask] Do you have sore throat, runny nose, or nasal congestion?
Patient: Yes, I have nasal congestion and a clear runny nose.
Doctor: [ask] Do you have any other symptoms such as wheezing, fatigue, or muscle aches?
Patient: No, I don't have any other symptoms.
Doctor: [inform] Based on your description, the most likely diagnosis is URTI.
```

Figure 3: The initial prompt includes the instruction I, the shots S, and the input D. The generated question $q_i$ of the prompted model (i.e., the DSAD) and the answer $a_i$ from the patient bot is presented in the remaining text in black.

## A.2 An Example of DR-CoT prompting

```
Act as a doctor taking a medical history from a patient. First review the patient's positive and negative
clinical findings, formulate ranked differential diagnosis, and ask comprehensive and important questions to
narrow down the differential diagnosis. Finally, I will make an accurate final diagnosis based on the collected
clinical findings. The diagnosis should be selected from the possible diagnoses.

Possible diagnoses: Bronchiolitis, Pericarditis, Myocarditis, HIV (initial infection), Bronchospasm / acute
asthma exacerbation, Acute pulmonary edema, Pneumonia, Pancreatic neoplasm, Bronchitis, Atrial fibrillation,
Anaphylaxis, Panic attack, Sarcoidosis, Pulmonary neoplasm, Spontaneous rib fracture, Viral pharyngitis, Croup,
Possible NSTEMI / STEMI, Chagas, Cluster headache, Acute dystonic reactions, Scombroid food poisoning, Acute
rhinosinusitis, Anemia, Spontaneous pneumothorax, Guillain-Barré syndrome, Whooping cough, Myasthenia gravis,
SLE, Ebola, Pulmonary embolism, Boerhaave, Unstable angina, Acute otitis media, Stable angina, Inguinal hernia,
Epiglottitis, PSVT, Tuberculosis, Acute laryngitis, Acute COPD exacerbation / infection, Localized edema,
Larygospasm, Influenza, GERD, Bronchiectasis, Allergic sinusitis, Chronic rhinosinusitis, URTI

<History taking>
Doctor: [ask] How may I help you today?
Patient: I have a cough.
Doctor: [ask] What's your sex and age?
Patient: I am an 81-year-old female.
Doctor: [ask] Based on the evidence: Positive clinical findings: 'cough', the ranked differential diagnosis is
'Viral pharyngitis, URTI, Bronchospasm / acute asthma exacerbation, Allergic sinusitis, Tuberculosis,
Bronchiectasis, Chronic rhinosinusitis, Acute COPD exacerbation / infection, Acute laryngitis, Bronchitis, GERD,
Acute otitis media, Influenza, Acute rhinosinusitis, and Pneumonia'. To narrow down the differential diagnosis,
the next question to ask is [Question] Did you have fever?
Patient: Yes, I have a fever.
Doctor: [ask] Based on the evidence: Positive clinical findings: 'cough and fever', the ranked differential
diagnosis is 'URTI, Viral pharyngitis, Tuberculosis, Acute laryngitis, Influenza, Bronchitis, Acute otitis
media, Pneumonia, Acute rhinosinusitis, Ebola, Bronchiolitis'. To narrow down the differential diagnosis, the
next question to ask is [Question] Did you have dyspnea or difficulty breathing?
Patient: No.
Doctor: [ask] Based on the evidence: Positive clinical findings: 'cough and fever'. Negative clinical findings:
'dyspnea', the ranked differential diagnosis is 'URTI, Viral pharyngitis, Acute laryngitis, Influenza, Acute
otitis media, Tuberculosis, Bronchitis, Acute rhinosinusitis, Pneumonia, and Ebola'. To narrow down the
differential diagnosis, the next question to ask is [Question] Do you have sore throat, runny nose, or nasal
congestion?
Patient: Yes, I have a sore throat.
Doctor: [ask] Based on the evidence: Positive clinical findings: 'cough, fever, and sore throat'. Negative
clinical findings: 'dyspnea', the ranked differential diagnosis is 'URTI, Influenza, Bronchitis, and Ebola'. To
narrow down the differential diagnosis, the next question to ask is [Question] What's your body temperature?
Patient: My body temperature is not measured.
Doctor: [ask] Based on the evidence: Positive clinical findings: 'cough, fever, and sore throat'. Negative
clinical findings: 'dyspnea', the ranked differential diagnosis is 'URTI, Influenza, Bronchitis, and Ebola'. To
narrow down the differential diagnosis, the next question to ask is [Question] Do you have myalgia or muscle
soreness?
Patient: Yes, I have diffuse muscle pain.
Doctor: [ask] Based on the evidence: Positive clinical findings: 'cough, fever, sore throat, and myalgia'.
Negative clinical findings: 'dyspnea', the ranked differential diagnosis is 'URTI, Influenza, and Ebola'. To
narrow down the differential diagnosis, the next question to ask is [Question] Do you feel extremely fatigue?
Patient: Yes, I am so tired that I am unable to do my usual activities or I am stuck in my bed all day long.
Doctor: [inform] Based on the evidence: Positive clinical findings: 'cough, fever, sore throat, myalgia, and
extremely fatigue'. Negative clinical findings: 'dyspnea', the ranked differential diagnosis is 'Influenza and
URTI'. Since the patient is extremely fatigue, the most likely diagnosis is Influenza.

<History taking>
Doctor: [ask] How may I help you today?
Patient: I have a cough.
Doctor: [ask] What's your sex and age?
Patient: I am a 24-year-old female.
Doctor: [ask] Based on the evidence: Positive clinical findings: 'cough', the ranked differential diagnosis is
'Viral pharyngitis, URTI, Bronchospasm / acute asthma exacerbation, Allergic sinusitis, Tuberculosis,
Bronchiectasis, Chronic rhinosinusitis, Acute COPD exacerbation / infection, Acute laryngitis, Bronchitis, GERD,
Acute otitis media, Influenza, Acute rhinosinusitis, and Pneumonia'. To narrow down the differential diagnosis,
the next question to ask is [Question] Do you have shortness of breath?
Patient: No.
Doctor: [ask] Based on the evidence: Positive clinical findings: 'cough'. Negative clinical findings: 'dyspnea',
the ranked differential diagnosis is 'Viral pharyngitis, URTI, Allergic sinusitis, Chronic rhinosinusitis, Acute
laryngitis, GERD, Acute otitis media, Influenza, Acute rhinosinusitis, Tuberculosis, Bronchitis, Pancreatic
neoplasm, Bronchospasm / acute asthma exacerbation, Bronchiectasis, Acute COPD exacerbation / infection, and
Pneumonia'. To narrow down the differential diagnosis, the next question to ask is [Question] Do you have sore
throat, runny nose, or nasal congestion?
Patient: Yes, I have a sore throat.
Doctor: [ask] Based on the evidence: Positive clinical findings: 'cough and sore throat'. Negative clinical
findings: 'dyspnea, rhinorrhea, and nasal congestion', the ranked differential diagnosis is 'URTI, Influenza,
Bronchitis, Ebola, and GERD'. To narrow down the differential diagnosis, the next question to ask is [Question]
Do you have fever?
Patient: No.
Doctor: [ask] Based on the evidence: Positive clinical findings: 'cough and sore throat'. Negative clinical
findings: 'dyspnea, rhinorrhea, nasal congestion, and fever', the ranked differential diagnosis is 'URTI,
Bronchitis, Influenza, Ebola, and GERD'. To narrow down the differential diagnosis, the next question to ask is
[Question] Do you have a burning sensation that starts in your stomach then goes up into your throat, and can be
associated with a bitter taste in your mouth?
Patient: No.
Doctor: [ask] Based on the evidence: Positive clinical findings: 'cough and sore throat'. Negative clinical
findings: 'dyspnea, rhinorrhea, nasal congestion, fever, and burning sensation in the chest', the ranked
differential diagnosis is 'URTI, Bronchitis, Influenza, and Ebola'. To narrow down the differential diagnosis,
the next question to ask is [Question] Have you noticed any changes in your voice like hoarseness?
Patient: Yes, I have noticed that the tone of my voice has become deeper, softer or hoarse.
Doctor: [ask] Based on the evidence: Positive clinical findings: 'cough, sore throat, and voice change'.
Negative clinical findings: 'dyspnea, rhinorrhea, nasal congestion, fever, or burning sensation in the chest',
the ranked differential diagnosis is 'Acute laryngitis, Viral pharyngitis, and URTI'. To narrow down the
differential diagnosis, the next question to ask is [Question] Do you smoke?
Patient: Yes, I smoke cigarettes.
Doctor: [ask] Based on the evidence: Positive clinical findings: 'cough, sore throat, voice change, and the
habit of smoking'. Negative clinical findings: 'dyspnea, rhinorrhea, nasal congestion, fever, or burning
sensation in the chest', the ranked differential diagnosis is 'Acute laryngitis, Viral pharyngitis, and URTI'.
To narrow down the differential diagnosis, the next question to ask is [Question] Have you been in contact with
people with similar symptoms in the past 2 weeks?
Patient: Yes, I have had a cold in the last 2 weeks.
Doctor: [inform] Based on the evidence: Positive clinical findings: 'cough, sore throat, voice change, the habit
of smoking, and the history of common cold in the past 2 weeks'. Negative clinical findings: 'dyspnea,
rhinorrhea, nasal congestion, fever, or burning sensation in the chest', the ranked differential diagnosis is
'Acute laryngitis, Viral pharyngitis, or URTI'. Since the patient has voice change and the history of common
cold in the past 2 weeks, the most likely diagnosis is Acute laryngitis.
```

```
<History taking>
Doctor: [ask] How may I help you today?
Patient: I have a cough.
Doctor: [ask] What's your sex and age?
Patient: I am a 44-year-old male.
Doctor: [ask] Based on the evidence: Positive clinical findings: 'cough', the ranked differential diagnosis is
'Viral pharyngitis, URTI, Bronchospasm / acute asthma exacerbation, Allergic sinusitis, Tuberculosis,
Bronchiectasis, Chronic rhinosinusitis, Acute COPD exacerbation / infection, Acute laryngitis, Bronchitis, GERD,
Acute otitis media, Influenza, Acute rhinosinusitis, and Pneumonia'. To narrow down the differential diagnosis,
the next question to ask is [Question] Do you have shortness of breath?
Patient: No.
Doctor: [ask] Based on the evidence: Positive clinical findings: 'cough'. Negative clinical findings: 'dyspnea',
the ranked differential diagnosis is 'Viral pharyngitis, URTI, Allergic sinusitis, Chronic rhinosinusitis, Acute
laryngitis, GERD, Acute otitis media, Influenza, Acute rhinosinusitis, Tuberculosis, Bronchitis, Pancreatic
neoplasm, Bronchospasm / acute asthma exacerbation, Bronchiectasis, Acute COPD exacerbation / infection, and
Pneumonia'. To narrow down the differential diagnosis, the next question to ask is [Question] Do you have chest
pain or tightness?
Patient: No.
Doctor: [ask] Based on the evidence: Positive clinical findings: 'cough'. Negative clinical findings: 'dyspnea,
chest pain, and tightness', the ranked differential diagnosis is 'Viral pharyngitis, URTI, Allergic sinusitis,
Chronic rhinosinusitis, Acute laryngitis, GERD, Acute otitis media, Influenza, Acute rhinosinusitis,
Tuberculosis, Bronchitis, Pancreatic neoplasm, Bronchospasm / acute asthma exacerbation, Bronchiectasis, Acute
COPD exacerbation / infection, and Pneumonia'. To narrow down the differential diagnosis, the next question to
ask is [Question] Do you have sore throat, runny nose, or nasal congestion?
Patient: Yes, I have nasal congestion and a clear runny nose.
Doctor: [ask] Based on the evidence: Positive clinical findings: 'cough, nasal congestion, and runny nose'.
Negative clinical findings: 'dyspnea, chest pain, and tightness', the ranked differential diagnosis is 'URTI,
Allergic sinusitis, Chronic rhinosinusitis, Acute rhinosinusitis, Influenza, Bronchitis, and
Bronchospasm / acute asthma exacerbation'. To narrow down the differential diagnosis, the next question to ask
is [Question] Do you have fever?
Patient: No.
Doctor: [ask] Based on the evidence: Positive clinical findings: 'cough, nasal congestion, and runny nose'.
Negative clinical findings: 'dyspnea, chest pain, tightness, and fever', the ranked differential diagnosis is
'URTI, Allergic sinusitis, Chronic rhinosinusitis, Acute rhinosinusitis, Influenza, Bronchitis, and
Tuberculosis'. To narrow down the differential diagnosis, the next question to ask is [Question] Do you have a
history of allergies or asthma?
Patient: Yes, I have close family members who suffer from allergies (any type), hay fever, or eczema. I also
have asthma or have used a bronchodilator in the past.
Doctor: [ask] Based on the evidence: Positive clinical findings: 'cough, nasal congestion, runny nose, and a
history of allergies or asthma'. Negative clinical findings: 'dyspnea, chest pain, tightness, and fever', the
ranked differential diagnosis is 'Allergic sinusitis, URTI, Chronic rhinosinusitis, Acute rhinosinusitis,
Bronchitis, and Bronchospasm / acute asthma exacerbation'. To narrow down the differential diagnosis, the next
question to ask is [Question] Do you have a history of exposure to environmental irritants such as smoke, dust,
or pollen?
Patient: Yes, I live in a big city and I am more likely to develop common allergies than the general population.
I have also traveled out of the country to AmerN in the last 4 weeks.
Doctor: [ask] Based on the evidence: Positive clinical findings: 'cough, nasal congestion, runny nose, a history
of allergies or asthma, and a history of exposure to environmental irritants'. Negative clinical findings:
'dyspnea, chest pain, tightness, and fever', the ranked differential diagnosis is 'Allergic sinusitis, URTI,
Chronic rhinosinusitis, Acute rhinosinusitis, Bronchitis, and Bronchospasm / acute asthma exacerbation'. To
narrow down the differential diagnosis, the next question to ask is [Question] Do you have severe itching in one
or both eyes?
Patient: Yes, I have severe itching in one or both eyes.
Doctor: [ask] Based on the evidence: Positive clinical findings: 'cough, nasal congestion, runny nose, a history
of allergies or asthma, a history of exposure to environmental irritants, and severe itching in one or both
eyes'. Negative clinical findings: 'dyspnea, chest pain, tightness, and fever', the ranked differential
diagnosis is 'Allergic sinusitis, URTI, Chronic rhinosinusitis, Acute rhinosinusitis, and Bronchospasm / acute
asthma exacerbation'. To narrow down the differential diagnosis, the next question to ask is [Question] Do you
have sneezing or nasal congestion?
Patient: Yes, I have nasal congestion and a clear runny nose.
Doctor: [inform] Based on the evidence: Positive clinical findings: 'cough, nasal congestion, runny nose, a
history of allergies or asthma, a history of exposure to environmental irritants, severe itching in one or both
eyes, and sneezing or nasal congestion'. Negative clinical findings: 'dyspnea, chest pain, tightness, and
fever', the ranked differential diagnosis is 'Allergic sinusitis, URTI, Chronic rhinosinusitis, and Acute
rhinosinusitis'. Since the patient has a history of allergies or asthma, a history of exposure to environmental
irritants, severe itching in one or both eyes, and sneezing or nasal congestion, the most likely diagnosis is
Allergic sinusitis.
```

Figure 4: The initial prompt includes the instruction I, the shots S, and the input D. The generated question $q_i$ of the prompted model (i.e., the DSAD) and the answer $a_i$ from the patient bot is presented in the remaining text in black. We invite a physician to write the DDx part of the two DR-CoT exemplars.

## A.3 AN EXAMPLE OF PATIENT BOT PROMPTING

```
Act as a patient consulting with a doctor during history taking, and respond to the doctor's question according
to the following background information.

<Background information>
Sex: Male, Age: 47
- I feel pain.
- The pain is:
* tugging
* burning
- The pain locations are:
* back of head
* top of the head
* temple(R)
- On a scale of 0-10, the pain intensity is 4
- On a scale of 0-10, the pain's location precision is 8
- On a scale of 0-10, the pace at which the pain appear is 5
- I feel like I am about to faint.
- I feel lightheaded and dizzy.
- I feel palpitations.
- I feel anxious.
- I regularly drink coffee or tea.
- I regularly consume energy drinks.
- I regularly take stimulant drugs.
- I have recently taken decongestants or substances that may have stimulant effects

<History taking>
Doctor: How may I help you today?
Patient: I feel palpitations.
Doctor: What's your sex and age?
Patient: I am a 47-year-old male.
Doctor: Do you feel pain?
Patient: Yes.
Doctor: Do you feel depressed?
Patient: No.
Doctor: How does your pain feel?
Patient: The pain is tugging and burning
Doctor: Where is your pain located?
Patient: The pain is located at the back of my head, the top of my head, and my right temple.
Doctor: Do you have nausea or vomiting?
Patient: No.
Doctor: Do you have anxiety?
Patient: Yes.
Doctor: How intense is the pain?
Patient: On a scale of 0-10, the pain intensity is 4.
Doctor: Do you feel lightheaded and dizzy?
Patient: Yes.
Doctor: Have you had chills or shivers?
Patient: No.
Doctor: Have you noticed light red blood or blood clots in your stool?
Patient: No.

<Background information>
Sex: Male, Age: 44
- I have close family members who suffer from allergies (any type), hay fever, or eczema.
- I have asthma or have used a bronchodilator in the past.
- My nose or the back of my throat is itchy.
- I have severe itching in one or both eyes.
- I have nasal congestion or a clear runny nose.
- I have a cough.
- I live in a big city.
- I am more likely to develop common allergies than the general population.
- I have traveled out of the country to AmerN in the last 4 weeks.

<History taking>
Doctor: How may I help you today?
Patient: I have a cough.
Doctor: What's your sex and age?
Patient: I am a 44-year-old male.
Doctor: Do you have shortness of breath?
Patient: No.
Doctor: Do you have chest pain or tightness?
Patient: No.
Doctor: Do you have sore throat, runny nose, or nasal congestion?
Patient: Yes, I have nasal congestion and a clear runny nose.
Doctor: Do you have fever?
Patient: No.
Doctor: Do you have a history of allergies or asthma?
Patient: Yes, I have close family members who suffer from allergies (any type), hay fever, or eczema. I also
have asthma or have used a bronchodilator in the past.
Doctor: Do you have a history of exposure to environmental irritants such as smoke, dust, or pollen?
Patient: Yes, I live in a big city and I am more likely to develop common allergies than the general population.
I have also traveled out of the country to AmerN in the last 4 weeks.
Doctor: Do you have severe itching in one or both eyes?
Patient: Yes, I have severe itching in one or both eyes.
Doctor: Do you have sneezing or nasal congestion?
Patient: Yes, I have nasal congestion and a clear runny nose.
```

Figure 5: The initial prompt includes the instruction I, the shot S, and the input P and D. The generated answer $a_i$ of the prompted model (i.e., the patient bot) and the inquired question $q_i$ from the DSAD is presented in the remaining text in black.

## B    DETAILS OF EXPERIMENTAL SETUPS

### B.1    DATASET

**Details of DDXPlus.** Compared with other AD datasets with only 4~5 diagnoses and 40~70 types of evidence (Wei et al., 2018; Xu et al., 2019), the DDXPlus dataset includes 49 diagnoses and 223 types of evidence. Each instance of DDXPlus (Tchango et al., 2022) represents a *patient*, and we use the following attributes of the patient for our experiments, : AGE, SEX, INITIAL_EVIDENCE (IE), EVIDENCES (i.e., the symptoms or other clinical findings experienced by the patient), and PATHOLOGY. The AGE, SEX, IE, and EVIDENCES are used to construct the patient profile P based on a rule-based template, and the PATHOLOGY is the ground truth diagnosis for computing the diagnostic accuracy. At the beginning of each dialogue session, AGE, SEX, and IE are provided to the doctor model to kick-start the dialogue.

**In-domain and Out-domain.** We refer to the test samples that share the same IE with our two exemplars as *in-domain*, and the ones that do not as *out-domain*. The rationale behind this notion is that the LLM only sees in-domain exemplars' IE, thus, other IEs are considered to be out-domain. For the selection of in-domain (ID) and out-domain (OD) IE, we choose *cough*, *dyspnea*, and *runny nose* (i.e., *toux*, *dyspn*, and *rhino_clair* in DDXPlus) as the IEs for ID, OD1, and OD2 respectively. These three symptoms is the most frequently occurred IE in the testset of DDXPlus with 618, 11305, and 6270 instances, respectively. As accessing LLMs is costly, especially with our limited budget, we randomly sample 200, 100, and 100 instances for ID, OD1, and OD2 to conduct our experiments.

### B.2    MODEL

For the model choice, we adopt the text-davinci-003 version of the InstructGPT via OpenAI API[2]. To the best of our knowledge, it is the most capable version of publicly available GPT-3 at the time we conduct our experiments. As for hyperparameters, we set the maximum tokens to 384 and the temperature to 0 (since diversity is not of necessity for our task). The remaining hyperparameters are kept as the default values.

### B.3    DIALOGUE TURN

To evaluate the diagnosis accuracy of dialogue turn $T$, if the final diagnosis has not been generated, we instruct the DSAD to make the final diagnosis by inserting a prefix "[inform]" in turn $T$'s prompt (see Appendix A.1, A.2). And to avoid lengthy dialogue, we set a pre-defined $T_{max}$ as the maximum number of turn allowed for making the final diagnosis. We determine $T_{max}$ by inviting a physician to interact with our patient bot, and find that 6~8 turns are required on average. Thus, we set $T_{max} = 8$ for our experiments. Similarly, if the DSAD fails to give a diagnosis before the $8$-$th$ turn, the prefix "[inform]" will be given to the DSAD to force the prediction.

## C    HUMAN EVALUATION

One of our hypotheses is that DR-CoT guides LLMs to generate a better $q_i$ for collecting crucial evidence, which ultimately leads to a more accurate final diagnosis. Establishing an accurate final diagnosis depends on two abilities: (1) asking critical questions to collect evidence, and (2) predicting the diagnosis based on the collected evidence. To verify that DR-CoT prompted LLMs indeed ask better questions, we invite a physician to evaluate the dialogues as there is no ground truth for $q_i$. During the evaluation, only the $(q_i, a_i)$ pairs are revealed to anonymize the promptimg methods. Table 1 summarizes the results, which show that DR-CoT prompting enables LLMs to ask questions which are more critical for establishing the correct diagnosis.

---

[2]https://platform.openai.com/docs/models/gpt-3

Table 1: Number of dialogues where the $q_i$s are deemed more critical for diagnosing the patient.

| Method | ID | OD1 | OD2 |
|---|---|---|---|
| Standard | 2 | 2 | 1 |
| DR-CoT | 18 | 8 | 9 |
| Total | 20 | 10 | 10 |

