# OpenReview forum: "Large Language Models Perform Diagnostic Reasoning"
_ICLR.cc/2023/TinyPapers — Submitted to Tiny Papers @ ICLR 2023_

### Official Review · Reviewer_PjD8 · 2023-03-29

**Confidence:** 4

**Summary Of Contributions:**

This paper extends chain-of-thought (CoT) prompting to medical reasoning for automatic diagnosis. The method is simple but effective, by mimicking the process of forming differential diagnosis (DDx) in patient-doctor interaction. It also proposes a reliable role-play framework for automatic evaluation, and shows that DR-CoT performs significantly better than standard CoT on this testbed.

**Rating:**

High Potential (HP): a submission which meets the reviewing criteria and has potential to make an impact on the field

**Strengths And Weaknesses:**

Strengths

1. Extend CoT prompting of Large Language Models to a new but important area, automatic diagnosis beyond conventional arithmetic, commonsense, and symbolic reasoning tasks.

2. This method is inspired by real patient-doctor interaction, which is explainable, and effectively leverages various abilities of LLMs like retrieval and summarization.

3. The method is robust since the improvements on out-domain test sets are more significant.

4. The language-model-role-play interactive evaluation framework is quite novel. The idea can be expanded to a wider range of application scenarios, and the author also shows its reliability through expert assessment.

Weaknesses

It’s better to compare DR-COT with some other baseline models on DDXPlus to see if it is good enough. As I see, the accuracy is only 20-30, which seems to be dangerous for a medical diagnosis.


**Suggested Changes:**

1. In Figure 1, the meaning of the x-axis, i.e., # of dialogue turns, is vague. More details are needed to be explained.

2. Better to put Appendix C into the main body.

---

> ### Author Response · Authors · 2023-05-29
> **Reply to reviewer PjD8**
>
> Thank you for the positive comments.
>
> **Regarding the Weaknesses:**
>
> > It’s better to compare DR-COT with some other baseline models on DDXPlus to see if it is good enough.
>
> Most existing baselines (e.g., DDXPlus) are fine-tuned on large-scale training data and only focus on the label prediction of AD without possessing the conversational ability of our proposed DR-CoT Bot.
> As the scope of this work focuses on how to elicit the diagnostic reasoning ability of LLM in few-shot, comparison with other baselines is not investigated, and is left as our future work.
>
> > As I see, the accuracy is only 20-30, which seems to be dangerous for a medical diagnosis.
>
> We acknowledge that solely applying DR-CoT to existing LLMs is not directly applicable considering the current performance.
> Instead, we hope the empirical improvement demonstrated by DR-CoT can serve as a stepping stone to facilitate research on medical reasoning and automatic diagnosis using LLMs.
> Thank you for the insightful feedback.
>
> **Regarding the suggested changes:**
>
> > In Figure 1, the meaning of the x-axis, i.e., # of dialogue turns, is vague. More details are needed to be explained.
>
> The “dialogue turn” mentioned in figure 1 refers to the turn of conversation on which we instruct the model to make the final diagnosis, i.e., by the “[inform]” prefix discussed in Appendix B.3.
> We acknowledge these details are not well-provided, and we have encompassed them in the result section of our final version.
> Thank you for bringing this to our attention.
>
> > Better to put Appendix C into the main body.
>
> We have added relevant information from Appendix C into the main content of the camera-ready version.
> Thank you for the constructive advice for improving the completeness of our work.

---

### Official Review · Reviewer_FH9i · 2023-04-01

**Confidence:** 4

**Summary Of Contributions:**

This paper proposes a new framework for improving the performance of dialogue systems in automatic diagnosis (AD), which requires knowledge-intensive, multi-step diagnostic reasoning. The authors show that their conversation system with DR-CoT prompting performs 15% better on diagnostic accuracy than the conventional prompt.

**Rating:**

High Impact (HI): a submission which meets the reviewing criteria and is predicted to make an impact on the field

**Strengths And Weaknesses:**

STRENGTHS
The paper proposes a novel approach to improving the performance of dialogue systems in automatic diagnosis by extending the Chain of Thoughts (CoT) approach to medical reasoning and incorporating a Diagnostic Reasoning (DR-CoT).
The authors propose a few-shot LLM-based dialogue system for AD, which in principle has an unbounded diagnosis label set.
The authors evaluate their proposed framework using a language-model-role-playing evaluation framework that simulates realistic patient-doctor interactions.

WEAKNESS
The paper should provide a more detailed explanation of the evaluation metrics used for measuring the performance of the dialogue system with DR-CoT prompting

**Suggested Changes:**

I suggest that, rather than putting key explanations into the appendix making the reader have to go to the bottom to get the broad clear concept, it can be summarized in the result and conclusion section while potentially reducing the wordings under the introduction section. This, I feel is because the author is constrained to a two-page limit. It can be better

---

> ### Author Response · Authors · 2023-05-29
> **Reply to reviewer FH9i**
>
> Thank you for the recognition of our work.
>
> **Regarding the weaknesses:**
>
> > The paper should provide a more detailed explanation of the evaluation metrics used for measuring the performance of the dialogue system with DR-CoT prompting
>
> Thank you for bringing this to our attention.
> The evaluation metrics used for measuring the performance of the dialogue system is the accuracy of the final diagnosis.
> Concretely, to evaluate the accuracy of dialogue turn $T$ (Figure 1), we instruct the model to make the final diagnosis by inserting the “[inform]” prefix (discussed in Appendix B.3) in turn $T$’s prompt.
> We have provided the above-described details in the result section of the camera-ready version.
>
> **Regarding the suggested changes:**
>
> > I suggest that, rather than putting key explanations into the appendix making the reader have to go to the bottom to get the broad clear concept, it can be summarized in the result and conclusion section while potentially reducing the wordings under the introduction section. This, I feel is because the author is constrained to a two-page limit. It can be better
>
> It is indeed due to the page limit that we are unable to provide the comprehensive details in the main content.
> We have added all necessary information and key explanation from the appendix into the result section of final version.
> Thank you for the constructive feedback.

---

### Official Review · Reviewer_NsA3 · 2023-04-01

**Confidence:** 4

**Summary Of Contributions:**

In this submission, the authors apply chain-of-thought prompting for medical diagnostics. They evaluate on DDXPlus, where they achieve a significant improvement in accuracy.

**Rating:**

High Potential (HP): a submission which meets the reviewing criteria and has potential to make an impact on the field

**Strengths And Weaknesses:**

Strengths:
- Automatic diagnosis is an important and intestine field of study, and important to increase accuracy as much as possible
- The experiments are a good indication that chain-of-thought prompting can have a significantly impact in reasoning in medical diagnosis
- Prompts are available in the appendix
- clear presentation

Weaknesses:
- The experiments are limited. It is not clear if the chain-of-thought prompting is the sole reason for the improvement or the choice of examples.
- It is not clear what “we invite a physician” means. Have they been paid? Is it an author?


**Suggested Changes:**

An experiment showing that the claim (of applying cot is useful) is robust when considering a different set of examples is missing.
A little more information on the evaluation is needed (see weaknesses above).

---

> ### Author Response · Authors · 2023-05-29
> **Reply to reviewer NsA3**
>
> Thank you for the positive comments.
>
> **Regarding the weakness:**
>
> > The experiments are limited. It is not clear if the chain-of-thought prompting is the sole reason for the improvement or the choice of examples.
>
> To isolate possible confounders, the prompts for our experiments are designed to only differ in the presence of the diagnostic reasoning process (the text colored in red in Section 2).
> Specifically, the questions asked by the doctor in the prompt of standard prompting are exactly the same as in DR-CoT prompting.
> We acknowledge that more rigorous experiments would further strengthen our proposed approach, thank you for the constructive feedback.
>
> > It is not clear what “we invite a physician” means. Have they been paid? Is it an author?
>
> An annotation fee is provided and the physician is indeed one of the authors.
> We understand an independent rater would benefit the soundness of our results, however, all evaluation in our work is conducted fairly and in an unbiased manner.
> Thank you for the valuable feedback.
>
> **Regarding the suggested changes:**
>
> > An experiment showing that the claim (of applying cot is useful) is robust when considering a different set of examples is missing.
>
> The robustness of DR-CoT can be verified based on experimental results of the out-domain setting.
> To further demonstrate the benefit of applying DR-CoT,  in the future, we plan to construct different sets of DR-CoT exemplars with broader coverage for more experimental settings.

---

### Meta-Review · Area_Chair_N4s3 · 2023-04-05

**Recommendation:** Invite to present
**Confidence:** 5

**Metareview:**

This paper proposes a novel approach based on chain-of-thought prompting for automatic diagnosis, which is an important research field. The paper is generally clearly written. Most of the claims are well supported. The key concern is the lack of analysis on the robustness of CoT in the concerned task.

**Summary:**

The paper proposes a novel approach to improving the performance of dialogue systems in automatic diagnosis by extending the Chain of Thoughts (CoT) approach to medical reasoning. The reviewers generally agree that this work is of high potential. This work can be improved by a more thorough analysis and explanation.

**Comments And Feedback To The Authors:**



(i) Summarizing the key explanations from the appendix to the result and conclusion sections.

**Reason For Not Giving A Higher Recommendation:**

The three reviewers questioned the experiments. The authors may consider improving the paper based on the following aspects.

(i) The robustness of the chain-of-thought prompting, e.g., the number of prompting examples, and different exemplar sampling techniques.

(ii) A more detailed explanation of the evaluation metrics used for measuring the performance of the dialogue system with DR-CoT prompting

(iii) Comparison with more baselines. The diagnostic accuracy is only 20-30, which seems to be impractical for real-world applications.

**Reason For Not Giving A Lower Recommendation:**

N/A

---

### Decision · Program_Chairs · 2023-04-08

Invite to present